# Utilization of Monosaccharides by *Hungateiclostridium thermocellum* ATCC 27405 through Adaptive Evolution

**DOI:** 10.3390/microorganisms9071445

**Published:** 2021-07-04

**Authors:** Dung Minh Ha-Tran, Trinh Thi My Nguyen, Shou-Chen Lo, Chieh-Chen Huang

**Affiliations:** 1Molecular and Biological Agricultural Sciences Program, Taiwan International Graduate Program, Academia Sinica and National Chung Hsing University, Taipei 11529, Taiwan; hatranminhdung@gmail.com; 2Department of Life Sciences, National Chung Hsing University, Taichung 40227, Taiwan; mytrinhnguyen0410@gmail.com; 3Graduate Institute of Biotechnology, National Chung Hsing University, Taichung 40227, Taiwan; 4Innovation and Development Center of Sustainable Agriculture, National Chung Hsing University, Taichung 40227, Taiwan

**Keywords:** *Hungateiclostridium thermocellum*, *Clostridium thermocellum*, *Ruminiclostridium thermocellum*, adaptive laboratory evolution, RNA-seq, whole genome sequencing, cellulosomal genes, EMP pathway, monosaccharides, hexose sugars

## Abstract

*Hungateiclostridium thermocellum* ATCC 27405 is a promising bacterium for consolidated bioprocessing with a robust ability to degrade lignocellulosic biomass through a multienzyme cellulosomal complex. The bacterium uses the released cellodextrins, glucose polymers of different lengths, as its primary carbon source and energy. In contrast, the bacterium exhibits poor growth on monosaccharides such as fructose and glucose. This phenomenon raises many important questions concerning its glycolytic pathways and sugar transport systems. Until now, the detailed mechanisms of *H. thermocellum* adaptation to growth on hexose sugars have been relatively poorly explored. In this study, adaptive laboratory evolution was applied to train the bacterium in hexose sugars-based media, and genome resequencing was used to detect the genes that got mutated during adaptation period. RNA-seq data of the first culture growing on either fructose or glucose revealed that several glycolytic genes in the Embden–Mayerhof–Parnas pathway were expressed at lower levels in these cells than in cellobiose-grown cells. After seven consecutive transfer events on fructose and glucose (~42 generations for fructose-adapted cells and ~40 generations for glucose-adapted cells), several genes in the EMP glycolysis of the evolved strains increased the levels of mRNA expression, accompanied by a faster growth, a greater biomass yield, a higher ethanol titer than those in their parent strains. Genomic screening also revealed several mutation events in the genomes of the evolved strains, especially in those responsible for sugar transport and central carbon metabolism. Consequently, these genes could be applied as potential targets for further metabolic engineering to improve this bacterium for bio-industrial usage.

## 1. Introduction

*Hungateiclostridium thermocellum* (*Clostridium thermocellum*, *Ruminiclostridium thermocellum*, *Acetivibrio thermocellus*), a thermophilic, gram-positive, anaerobic bacterium, exhibits a strong capacity to efficiently degrade crystalline cellulose. *H. thermocellum* produces various industrially important fermentation products, such as ethanol, acetic acid, lactic acid, and hydrogen, thus understanding its responses to different carbon sources helps improve production of the desired end-products. To date, several studies focusing on the regulatory mechanisms of specific genes of interest [1,2,3,4,5,6,7] and genomic [8,9], transcriptomic [10,11,12,13,14], metabolomic [15,16], proteomic [17,18,19,20], and integrated omics [21,22,23,24] studies have been performed to expand our knowledge of *H. thermocellum* genetics, gene expression, metabolism, and physiology. However, most previous studies utilized the preferred substrates of *H. thermocellum,* such as cellobiose [9,21,24], crystalline cellulose [10,19], cellobiose and crystalline cellulose in combination [17,25,26], or pretreated switchgrass and *Populus* [12], pretreated yellow poplar and cellobiose [14]. The accumulation of glucose at the end of substrate fermentation was noted in a previous study [27] and the presence of glucose in medium, in turn, inhibits the catalytic activity of cellulosomes [28]. Accordingly, improving the utilization of glucose by *H. thermocellum* is beneficial for cellulosic bioethanol production. Although the chemical formula of glucose and fructose is the same, differences in their molecular formations leads to changes in cellulase production of glucose- and fructose-adapted cells. This phenomenon was first observed by Johnson et al. [29] decades ago, however, it has not been elucidated on a point of view of mRNA expression level. Since *H. thermocellum* always demonstrates impaired growth on comparatively simple monosaccharides such as glucose, fructose, or sorbitol [29,30,31], some research groups were interested in investigating the production of glycolytic enzymes [32], the production of cellulase [29], sugar transport systems [33,34], mutation of the bacterium [30], and carbon metabolism [31] in this bacterium when growing on these unfavored substrates. *H. thermocellum* experiences a long lag phase when growing on these substrates, which Nochur et al. [30] assumed might be due to the time needed for genetic changes in the bacterial genome to allow it to grow on fructose or glucose. However, these authors did not determine which genes got mutated during the extended lag period. In another study, Nochur et al. speculated that the long lag phase of fructose-adapted cells (FAs) and glucose-adapted cells (GAs) could be due to the lower intracellular pH in these cells than in cellobiose-grown cells (CGs) [31]. However, this could not explain the newly acquired characteristics of the evolved phenotypes, especially in terms of genetic and mRNA expression changes.

Adaptive laboratory evolution (ALE) has been widely used to better understand the basic mechanisms of molecular evolution and the genomic changes that accumulate in microbial populations during long-term selection under specific growth conditions [35]. With rapid advances in transcriptomic profiling and next-generation sequencing (NGS), phenotype–genotype correlations can be easily obtained [36]. In a previous ALE study, *H. thermocellum* was cultured in *Populus* hydrolysate, a medium that contains various compounds that are toxic to the bacterium. After 117 transfers from 5–17.5% (*v*/*v*) *Populus* hydrolysate, 73 mutations were identified [8]. These mutations were found to be related to cellular repair and energy metabolism, which helped the bacterium grow better than the wild type in the toxic medium [11]. A two-stage adaptive evolution was applied to train an engineered *H. thermocellum* AG553 (DSM1313 Δ*hpt* Δ*hydG* Δ*ldh* Δ*pfl* Δ*pta-ack*) for a faster growth and higher ethanol titer [37]. As a result, the resulting evolved strain LL1210 can produce 22.4 ± 1.4 g/L ethanol from 60 g/L cellulose. Recently, Holwerda et al. found that a mutation in the *adhE* gene induced by ALE allowed *H. thermocellum* to grow better and achieve higher ethanol yield [9]. In the present study, the cellobiose-grown cells were used as starting materials to inoculate into fructose-, glucose-, or cellobiose-containing medium. After the first culture event, the fructose-adapted cells (FAs1) and glucose-adapted cells (GAs1) were collected for RNA-seq, with the cellobiose-grown cells (CGs1) used as the control, to elucidate the responses of *H. thermocellum* to nutritional stress. The Embden–Meyerhof–Parnas (EMP) pathway was particularly emphasized since this is the predominant glycolytic route in this bacterium [16]. Subsequently, the ALE strategy was applied to improve growth rate and hexose sugars metabolism of FAs1 and GAs1. After seven serial transfers, genome resequencing was carried out to identify genetic changes that occurred in the genome of the evolved FAs8 and GAs8 strains. To date, only one recently published study by Yayo and coworkers [38] has used genomic analysis combined with reverse engineering to elucidate the mechanism of adaptation of *H. thermocellum* to hexose sugars. In the study of Yayo et al. [38], *H. thermocellum* DSM 1313 was cultured in carbon-limited chemostats with increasing glucose or fructose concentration and decreasing cellobiose concentration setting. Our approach, however, is quite different from their study, since *H. thermocellum* ATCC 27,405 was cultured in a static batch culture with sole fructose, glucose, or cellobiose added to the growth medium from the beginning of the fermentation. Our transcriptomic profiles of FAs1, FAs8, GAs1, GAs8, and CGs1 and the genome resequencing data of the evolved strains FAs8 and GAs8 aid to support and confirm the results of the previous studies, thereby shedding more light on the interesting abovementioned issues and broadening our knowledge in this area.

## 2. Materials and Methods

### 2.1. H. thermocellum ATCC 27,405 Growth Conditions and Medium Preparation

One liter of the modified GS-2 medium contained 1.5 g KH_2_PO_4_, 2.9 g K_2_HPO_4_, 3 g sodium citrate tribasic dihydrate (Na_3_C_6_H_5_O_7_), 2.1 g urea, 5 g 3-(N-morpholino) propanesulfonic acid (MOPS), 2 mg resazurin, and 1 g L-cysteine. A 10-fold trace salt solution contained 26 mg MgCl_2_, 11.3 mg CaCl_2_, and 0.125 mg FeSO_4_.7H_2_O. A 100-fold vitamin solution contained 2 mg pyridoxine hydrochloride, 0.2 mg biotin, 0.4 mg p-aminobenzoic acid and 0.2 mg vitamin B12 [39]. Trace salt and vitamin solutions were sterilized using a 0.22-µm filter (StarTech, Taiwan). All solutions were prepared with distilled water from an EcoQ Combo (LionBio, Taiwan). The pH of the GS-2 medium was adjusted to 7.2 using 5 M NaOH and purged extensively with pure nitrogen gas to create an anaerobic environment. GS-2 medium was autoclaved at 121 °C for 20 min, and 1% (*v*/*v*) trace salt solution and 1% (*v*/*v*) vitamin solution were added to the modified GS-2 when the medium was cooled to 50 °C.

To prepare the seed inoculum, *H. thermocellum* was grown in modified GS-2 medium supplemented with cellobiose (5 g/L) until the stationary phase was reached (OD_660_ ~ 0.7) and then inoculated 1% (*v*/*v*) into the same medium containing either glucose (10 g/L), fructose (10 g/L), or cellobiose (5 g/L). Three independent sets of CGs1, FAs1, and GAs1 were performed for growth pattern determination, gas and ethanol measurements, residual reducing sugar tests, and cellulosomal enzyme assays. Cells were grown in batch culture at 60 °C in 250-mL serum bottles containing 100 mL of modified GS-2 medium. Cell growth was monitored daily based on the measurement of optical density at A_660_ using a GeneQuant 1300 spectrophotometer (Biochrom, Holliston, MA, USA). The cell mass was quantified from the pellet protein as described by Zhang and Lynd [40], except that the Bradford assay, with bovine serum albumin (BSA) (Sigma-Aldrich, St. Louis, MI, USA) as the standard, instead of Peterson’s method. Briefly, 1 mL of culture broth was centrifuged using a Hitachi Tabletop centrifuge CT15RE (Hitachi Koki Co., Ltd., Tokyo, Japan) at 12,800× *g* for 10 min. The pellet was washed with distilled water once, and 1 mL of 0.2 M NaOH and 1% SDS were added to the pellet. The sample was incubated at 28 °C for 30 min with occasional stirring. The pellet was resuspended, and 0.25 mL of 0.8 M HCl was added and mixed well to neutralize the sample. After centrifugation at 12,800× *g* for 5 min, the protein in the supernatant was determined. One A_660_ unit corresponds to 0.85 g/L of cell biomass and carbon molarity in cell biomass was calculated based on the elemental composition of cells (C_5_H_8_O_2_N), which corresponds to a molecular weight of 114.12 g/mol [41].

### 2.2. Phosphoric Acid Swollen Cellulose and Cellulosome Isolation

Amorphous cellulose was prepared by phosphoric acid swelling as described by Zhang et al. [42]. Cellulosomes were isolated from 100 mL of cellobiose-, glucose- and fructose-fermentation broth collected at late stationary phase using the affinity digestion protocol described by St Brice et al. [43]. Briefly, cell-free broth was recovered by centrifuging the fermentation broth at 12,800× *g* for 40 min at 4 °C using a Hermle Z326K centrifuge (Benchmark, Wehingen, Germany). The supernatant was decanted, and the pH was adjusted to 7.0 with 2 M NaOH as evaluated with a S20 SevenEasy™ pH meter (Mettler-Toledo, PoMe, Australia). The cell-free broth was incubated with amorphous cellulose (10 mg/100 mL of cell-free broth) overnight at 4 °C to allow the binding of the cellulosomes to the cellulose. Amorphous cellulose with bound enzymes was centrifuged at 12,800× *g* for 30 min at 4 °C, and the pellet was resuspended in 20 mL dialysis buffer (50 mM Tris-base, 10 mM CaCl_2_, 5 mM dithiothreitol (DTT), pH 7.0). The amorphous cellulose suspension was then dialyzed in membrane bags (Spectrum^TM^ Spectra/POR^TM^1 MWCO 6-8000, Fisher Scientific, Hampton, NH, USA) at 55 °C against two liters of distilled water to initiate amorphous cellulose degradation by the enzyme. Distilled water was changed every 60 min to avoid inhibition of the cellulosomes by the degradation products. After 5 h of incubation, the transparent suspension was centrifuged at 3000× *g* for 30 min at 4 °C, decanted and stored at −20 °C for protein and activity analysis. Total protein of the extracted cellulosomes was measured with a Bio-Rad Bradford protein assay with BSA as a standard.

### 2.3. Avicelase, CMCase, and Xylanase Assays

The assays were performed as described by Zhang et al. [44]. Briefly, 4.1 mL of the well-suspended Avicel solution (2.44 g of dry Avicel/100 mL distilled water) and 0.5 mL of Tris-HCl buffer (0.5 M Tris; pH 7.0; 0.1 M CaCl_2_, and optional 1.5% (*w*/*v*) sodium azide (NaN_3_) were added into 16 × 125 mm anaerobic culture Hungate tubes. The tubes were sealed with 9-mm screw caps with butyl rubber stoppers, held under vacuum, and flushed with pure nitrogen at least 3 times for 5 min each. DTT at 0.5 M was added to the tubes (0.1 mL/tube) before the enzyme activity assay. The tubes were prewarmed in a water bath at 60 °C for 10 min, and 0.3 mL of the extracted cellulosome solution dilution series was added to the tubes. After the first 10 min of adsorption and reaction, 0.5 mL of sample (well-mixed Avicel suspension) was withdrawn as the starting point, followed by removal of an additional 0.5 mL every 60 min. The reactions were stopped by transferring the tubes to an ice water bath, and then the samples were placed in precooled 1.5-mL microcentrifuge tubes. The samples were centrifuged at 13,000× *g* for 3 min. The net soluble sugar released during the hydrolysis process was calculated by subtracting the amount of sugar measured at the starting point, and enzyme activity was determined based on the linear correlation between sugar released and enzyme concentration. In the CMCase assay, 2% (*w*/*v*) carboxymethylcellulose (CMC) was dissolved in citrate buffer (50 mM, pH 4.8). A volume of 0.5 mL of the diluted enzyme was added into the Hungate tubes. One half mL of the CMC solution was then added to the test tubes, and the tube contents were mixed well by pipetting. The mixture was incubated at 50 °C for 30 min, and 3 mL of DNS solution was added into the test tubes to stop the reaction. The Hungate test tubes were boiled for 5 min in boiling water and then placed in an ice water bath. The xylanase activity of purified cellulosomes was measured by the colorimetric assay described by Ribeiro et al. [45] using birchwood xylan as the substrate, and the reducing sugars were determined using the DNS method with xylose as a control. Briefly, the reaction mixture (0.05 mL substrate (1% *w*/*v*) in 50 mM sodium acetate buffer, pH 5, and 0.01 mL of enzyme solution) was incubated at 70 °C in a water bath for 5 min. The reaction was stopped by adding 0.1 mL of DNS and immediately boiling for 5 min. Quantification of the reducing sugars released as the result of enzyme activity was estimated by A_540_ measurements using a Beckman Coulter-PARADIGM™ microplate reader (Beckman Coulter, Chaska, MN, USA), where one unit of enzymatic activity was defined as the amount of the extracted cellulosomes that produced 1 µmol/min of reducing sugars.

### 2.4. Measurement of Residual Reducing Sugar, Carbon Dioxide, and Ethanol

Reducing sugars were measured using the DNS method [46], and the standard curves were constructed using 5-fold serial dilutions of a pure chemical sample. The standard curves for glucose, fructose and cellobiose assays were built separately, as these sugars have different reactivities with the DNS reagent [47]. To prepare samples for ethanol and reducing sugar determination, 1 mL of fermentation broth was centrifuged at 12,800× *g* for 10 min at 4 °C. The pellet was discarded, and the supernatant was filtered using a 0.22-µm filter (StarTech, Taipei, Taiwan). CO_2_ was measured daily, but ethanol and reducing sugars were analyzed using samples that were taken at late stationary phase. These major fermentation end-products were determined by a GC Agilent 7890A (Agilent Technologies, Santa Clara, CA, USA) equipped with a J&W 122-3232: 30 m × 250 µm × 0.25 µm DB-FFAP column, with nitrogen as the carrier gas at a flow rate of 30 mL/min. For ethanol measurement, the front inlet was used, the detector was kept at 225 °C, and the oven was heated from 50 °C to 150 °C (50–100 °C at a ramp rate of 30 °C/min and 100–150 °C at a ramp rate of 20 °C/min). For CO_2_ measurement, the back inlet was used, the detector was kept at 225 °C, and the oven was operated isothermally at 50 °C.

### 2.5. RNA Isolation and Library Preparation and Sequencing

Two independent sets of CGs1, FAs1 and GAs1 from two biological replicates and one set of each FAs8 or GAs8 from the two biological replicates were harvested at the late stationary phase for RNA-seq as described in the Figure 1. All 4 sets of biological replicates were used for RT-qPCR of selected genes. Total RNA was extracted using TRIzol (R) Reagent (Invitrogen, Waltham, MA, USA) according to the instruction manual. Purified RNA was quantified at OD_260nm_ using an ND-1000 spectrophotometer (NanoDrop Technology, Wilmington, DE, USA) and quality checked using a Bioanalyzer 2100 instrument (Agilent Technologies, CA, USA) with an RNA 6000 LabChip kit (Agilent Technologies, CA, USA).

All RNA sample preparation procedures were carried out according to Illumina’s official protocol. Agilent’s SureSelect Strand-Specific RNA Library Preparation Kit was used for library construction, followed by size selection with AMPure XP beads (Beckman Coulter, Carlsbad, CA, USA). The sequences were determined using Illumina sequencing-by-synthesis (SBS) technology (Illumina, CA, USA). Sequencing data (FASTQ reads) were generated based on the Illumina Conversion program bcl2fastq v2.20. The raw counts of FAs1, GAs1, CGs1 were presented in Appendix A, and those of FAs8, GAs8, and CGs1 were shown in Appendix A.

### 2.6. Differential Gene Expression Analysis

All RNA-seq data analyses and graphics production were performed using R (ver. 3.6.1). The expression levels were calculated as reads per kilobase per million reads (RPKM). The Bioconductor package edgeR (ver. 3.28.1) [48] was used for differential expression analysis. The edgeR was built based on a parametric method that performs DE analysis based on the assumption that the inputted raw data follow a negative binomial distribution. Genes with significant low expression levels (raw count < 3 or RPKM value < 0.3) in the treated samples (FAs1 and GAs1) and the control sample (CGs1) were excluded. Genes with ≥2-fold change or ≤0.5-fold change (log_2_-fold change ≥ 1 or log2-fold change ≤ −1) and a false discovery rate (FDR) of *p* < 0.01 were considered significantly differentially expressed.

### 2.7. Sample Preparation for Genomic Analysis

The evolved populations that had been serially passaged on fructose and glucose were used for genome resequencing. Each transfer included about 100-fold dilution (1%, *v*/*v*), or approximately 5 generations. Two independent sets from two biological replicates of the evolved strains FAs8 or Gas8 (8th transfer ~ 42 generations for FAs8 and ~40 generations for GAs8), which were considered the endpoint strains, were harvested, and stored at −80 °C. Genomic DNA was extracted by a WelPrep DNA kit (Welgene Biotech, Taipei, Taiwan) according to the manufacturer’s instructions. DNA samples with OD 260/280 in the range of 1.8~2.0 were quantity/quality assessed using the Agilent Genomic DNA ScreenTape assay in conjunction with the 4200 TapeStation system (Agilent Technologies, CA, USA).

### 2.8. Library Construction and Sequencing

Total DNA (10 µg) was sonicated using a Covaris M220 focused ultrasonicator (Covaris, Woburn, MA, USA) to sizes ranging from 400 to 500 bp. DNA sizing was checked using the Agilent D1000 ScreenTape assay with the 4200 TapeStation system. Approximately 0.5 μg fragmented DNA was end-repaired, A-tailed and adaptor-ligated following the Illumina TruSeq DNA preparation protocol. After library construction, samples were mixed with MiSeq Reagent Kit v3 (600 cycles) and loaded onto a MiSeq cartridge. Then, a 2 × 300 bp paired-end sequencing run was performed on the MiSeq sequencer platform (Illumina, San Diego, CA, USA).

### 2.9. Read Mapping, Variant Calling and Annotation

The raw sequences generated from the sequencer first went through a filtering process to obtain good quality reads. FASTP was implemented to trim bases or remove reads according to quality scores. The reference genome sequence and annotation GFF file were downloaded from the NCBI RefSeq genome database. The trimmed reads were aligned against the reference genome using the bwa aligner with the BWA-MEM algorithm. Then, the Picard tool MarkDuplicates was used to remove duplicate alignments. The GATK tool HaplotypeCaller was used for variant detection. Finally, snpEff was used to perform variant annotation.

## 3. Results and Discussion

### 3.1. Growth Patterns, Sugar Utilization, Cellulosomal Enzyme Activities and Ethanol Productivity of CGs1, FAs1, and GAs1

As expected, the culture of *H. thermocellum* ATCC 27,405 on hexose sugars took much time due to a defeat in central carbon metabolism and a lack of an appropriate monosaccharide transport system. After a long lag phase of adaptation, *H. thermocellum* eventually used glucose or fructose as a sole carbon source and energy for its growth (Figure 2A). This finding is in agreement with the results of Hernandez [33], Johnson et al. [29], and Yayo et al. [38], which showed that the *H. thermocellum* bacterium could grow on fructose and glucose only after an extended lag time. However, the final cell biomass of FAs1 and GAs1 was still lower than that of CGs1, in contrast to the results obtained in the work of Johnson et al. [29], who found that the final *H. thermocellum* cell density of FAs1, GAs1 and CGs1 was comparable. As the most powerful hydrolytic machinery that has been identified in a microorganism to date, *H. thermocellum* cellulosomal function, composition, and regulation patterns have been a popular research topic for decades [1,2,3,4,7,12,14,17,18,25,26,28,29,43,49,50,51,52,53,54,55,56,57]. Regarding cellulosomal enzyme activities, FAs1 cellulosomes expressed the highest exoglucanase, endoglucanase, and xylanase activities, followed by CGs1 cellulosomes (Figure 2B). Surprisingly, the lowest cellulosomal enzyme activities were found in GAs1, even though *H. thermocellum* growing on glucose is not subject to carbon catabolite repression [29]. The observed GAs1 enzyme activities were in agreement with a study by Zhang et al. [28], who used kinetic models and empirical experiments to prove that glucose was an inhibitor of the *H. thermocellum* cellulosome. Other reports also indicated that cellobiose was a strong inhibitor of cellulosomes and cellulases [53,58]. These previous studies could partly explain the low activities of cellulosomal enzymes derived from CGs1 and GAs1 compared to those derived from FAs1. As fructose and glucose are unfavored carbon sources for *H. thermocellum* growth, it is not surprising that their consumption was limited, representing large amounts of fructose and glucose remained in the culture broths at the end of the fermentation. Cellobiose, in contrast, is the preferred carbon source of *H. thermocellum*, thus, this disaccharide was rapidly consumed by the bacterium, enabling robust growth and an effective end-product formation. Consequently, the ethanol productivity, sugar consumption, and cell biomass of CGs1 were significantly greater than those of FAs1 and GAs1 (Figure 2C).

### 3.2. Adaptive Laboratory Evolution

In the beginning, the cellobiose-grown cells at exponential phase (OD_660_ ~ 0.6) were inoculated (1%, *v*/*v*) into fresh fructose- or glucose-based medium. A dark yellow color appeared in the FAs1 fermentation broth, simultaneously with some motile granules in the medium, suggesting a stress response of the organism to fructose. Similarly, the yellowish color also appeared in the GAs1 fermentation broth. However, the color soon became light yellow and then milky white after some serial transfers. From the first to the seventh transfer, the duration of the lag phase gradually decreased (from approximately 168 ± 22 h for FAs1 and 216 ± 51 h for GAs1 to less than 12 h for both FAs8 and GAs8). After some transfers, the bacterial biomass of the intermediate strains constantly increased relative to that of FAs1, along with an improved ability to consume fructose, as shown in Table 1. Consequently, the ethanol yield of the end-point strain FAs8 was comparable with that of the parent strain FAs1, but it exhibited a dramatically improved growth rate and a significantly greater biomass relative to FAs1 (Table 1).

A similar pattern was observed in the intermediate strain GAs, reflected in a greater biomass and faster glucose consumption compared to the first culture GAs1 (Table 1). We observed a longer lag phase on glucose feeding than that on fructose, whereas the opposite observation was recorded by Yayo et al. [38], with a 42 h-lag phase for glucose-grown cells and a longer than 80-h-lag phase for fructose-grown cells. The higher cell biomass on fructose rather than on glucose observed in our study was in accordance with Yayo and colleagues’ finding [38]. Glucose consumption during fermentation and the yield of ethanol, however, were lower than those of FAs8 (Table 1).

### 3.3. Growth of FAs8 on Glucose-Based Medium and Growth of GAs8 on Fructose-Based Medium

The reduced lag phase was noticed when the evolved strain FAs8 was inoculated (1%, *v*/*v*) into a fresh glucose-containing medium and the evolved strain GAs8 was inoculated into a fresh fructose-containing medium (Figure 3). This finding indicated that the evolved FAs8 can properly uptake and ferment glucose and vice versa, which confirms that the acquired mutation rather than a physiological adaptation promotes its rapid growth on monosaccharides, regardless of the type of hexose sugars. Furthermore, the biomass of GAs8 on fructose (GAs8-Fructose) was also higher than that of FAs8 on glucose (FAs8-Glucose), suggesting fructose is a more favored hexose sugar than glucose for the evolved *H. thermocellum*.

In addition, to determine whether the FAs8 and GAs8 strains still conserve the capacity for consuming cellobiose, the evolved strains were separately inoculated (1%, *v*/*v*) into a fresh cellobiose-containing medium. As a result, these evolved strains can grow effectively on cellobiose, with a slightly higher specific growth rate for GAs8 (0.069 ± 0.003 h^–1^) relative to FAs8 (0.061 ± 0.002 h^–1^), in agreement with the results in Yayo et al. [38]. These results support an assumption that beneficial genetic changes are needed for better growth on monosaccharides, while the capability to ferment cellobiose was appropriately maintained in the evolved strains.

### 3.4. Transcriptomic Profiling of CGs1, FAs1, GAs1 and Their Evolved Strains FAs8 and GAs8

The expression patterns of cellulosomal genes in RNA-seq aided in confirming the results of enzyme activities. Our RNA-seq data showed that many carbohydrate-active enzyme (CAZyme)-encoding genes in GAs1 remained at low expression levels, while these genes exhibited higher expression levels in FAs1 (Appendix A). To regulate the expression of cellulosomal genes in response to different substrates (i.e., cellulose, pectin, or xylan), *H. thermocellum* has an alternative sigma/antisigma (SigI/RsgI) factor system that works harmoniously via an external carbohydrate-sensing mechanism [3,4,51]. Although the carbon sources used in the present study were not specific carbohydrate targets of the SigI/RsgI system, several pairs of SigI/RsgI factors were differentially expressed (Appendix A), which suggests potential roles in extracellular carbohydrate sensing of these antisigma factors in the presence of monosaccharides. In addition, given that the cellulosome could be inhibited by glucose [28] and cellobiose [59], latent regulators of non-cellulosomal enzymes might activate the expression of their regulons to hydrolyze any potential substrate in culture medium. Of the various non-cellulosomal genes known to date, we identified a dozen with differential expression in FAs1 and/or GAs1 (Appendix A). In the study of Johnson et al. [29], the poor growth of *H. thermocellum* on fructose or sorbitol was accompanied by a significant increase (five- or six-fold higher) in the specific production of cellulase. This result was in accordance with Dror et al. [4,5], who found that the transcription of the exoglucanase gene *celS* (Cthe_2089), scaffolding protein gene *cipA* (Cthe_3077), two anchoring scaffoldin genes *olpB* (Cthe_3078) and *orf2p* (Cthe_3079) were affected by the growth rate rather than by the substrate itself. In other words, a slower growth rate of *H. thermocellum* should be associated with higher cellulase activity. In our RNA-seq data, *celS*, *cipA*, *olpB*, and *orf2p* were highly expressed in FAs1 and GAs1, except *olpB* was downregulated in GAs1 (Appendix A). However, the production of specific cellulase was remarkably decreased when *H. thermocellum* cells were properly adapted to fructose [29], suggesting that carbon catabolite repression was responsible for the reduction of cellulase production. To further clarify this finding from a transcriptomic point of view, cell samples from FAs8 and GAs8 were collected for RNA-seq. The expression levels of key cellulosomal genes determined by RNA-seq and confirmed by RT-qPCR were dramatically decreased from FAs1 to FAs8, with CGs1 as the reference (Appendix A), in agreement with the enzymatic analysis of Johnson et al. [29]. In their study, Johnson and colleagues [29] found that after several numbers of transfers, the cellulase yields of fructose-adapted cells were even lower than those of strains grown on cellobiose or glucose. On the other hand, numerous cellulosomal genes, such as *celK* (Cthe_0412), *celB* (Cthe_0536), *celO* (Cthe_0625), acetyl xylan esterase (Cthe_0798), *celH* (Cthe_1472), *celS*, glycoside hydrolase family 11 (Cthe_2167), *celT* (Cthe_2812), *olpB*, and *orf2p*, were upregulated in GAs8 relative to those in CGs1 and GAs1 (Appendix A). This result suggests a de-repression of cellulase genes in the evolved GAs8 when glucose might be no longer an inhibitor of GAs8 cellulosomes. The list of primers used for RT-qPCR is shown in Appendix A, and the correlation between RNA-seq and RT-qPCR is presented in Appendix A.

### 3.5. EMP Pathway

#### 3.5.1. Genes Upstream of Phosphoenolpyruvate (PEP)

RNA-seq data in the present study demonstrated the dynamics of the transcriptome in the EMP pathway. This finding is in line with Xiong et al. [16], who found that the EMP pathway was the predominant glycolytic route in *H. thermocellum*. Many key glycolytic genes in the EMP pathway were highly upregulated in CGs1 compared to FAs1, FAs8, GAs1, and GAs8 (Figure 4). The log_2_ fold change of FAs1 and FAs8 relative to CGs1 and log_2_ fold change of GAs1 and GAs8 relative to CGs1 are presented as gene name (number1/number2|number3/number4), respectively. As might be expected, the gene Cthe_0275 encoding a cellobiose phosphorylase (CBP, EC 2.4.1.20) that catalyzes the phosphorolysis of cellobiose into glucose 1-phosphate (Glc1P) and glucose (Glc) was upregulated in CGs1 relative to FAs1, GAs1 and the evolved FAs8 and GAs8. However, another CBP gene (Cthe_1221) was upregulated in FAs1 and GAs1. The role of this CBP gene in fructose metabolism and the possible existence of a regulator for its elevated level of expression remains to be studied. From Glc to 2-phosphoglycerate (2PGA), various genes involved in the conversion of sugar phosphate intermediates were found to be upregulated in CGs1 relative to FAs1, GAs1, FAs8 and GAs8. Specifically, Cthe_2938 encodes glucokinase (GCK, EC 2.7.1.2) which converts Glc to glucose 6-phosphate (Glc6P), and was upregulated in CGs1, in congruence with its elevated expression in carbon-repleted medium [22]. In a previous study, Patni and Alexander [32] found that the increase in GCK activity in GAs was proportional to the increase in the yield of cell mass. They argued that GCK was likely an inducible enzyme, as its synthesis could be induced by glucose, fructose, or mannose. Intriguingly, the activity of GCK increased in mannitol-grown cells after their suspension in glucose medium, suggesting a fundamental role of GCK in glucose metabolism. This finding, however, was in contrast with the expression pattern of Cthe_2938 in GAs8 as its activity was decreased with respect to that of CGs1. According to Yayo et al. [38], the inactivation of gene Cthe_0390, a Repressor-ORF-Kinase (ROK) transcriptional regulator, may help *H. thermocellum* to metabolize glucose and fructose better, thereby ameliorating growth rate and biomass formation. In our study, the decreased expression of Cthe_0390 in FAs8 and GAs8 was also recorded, suggesting a disruption to transcriptional process in this regulator.

The elevated expression of the *GCK* gene in CGs1 could be explained by the abundance of intracellular Glc formed from cellobiose, which is easily taken up from the cellobiose culture and then undergoes phosphorolytic cleavage by CBP to produce Glc1P and Glc. The high accumulation of sugar phosphate intermediates in FAs and GAs, which was first described by Nuchur et al. [31], was attributed to the accumulation rate of Fru6P being much greater in fructose medium than in cellobiose medium. This finding suggested that *H. thermocellum* metabolizes cellobiose much faster than hexose sugars. The accumulation of cellobiose, Glc6P, Fru6P, and 3PGA found in *H. thermocellum* growing under ethanol stress [21] suggests that a disrupted central carbon metabolism might severely hinder sugar fermentation. This study also indicated that the phosphofructokinase (PFK)- and phosphoglycerate mutase (PGAM)-encoding genes were downregulated under ethanol stress conditions, in agreement with what was observed for FAs1 and GAs1 under nutritional stress in the present study (Figure 4).

Other genes, such as Cthe_0217 encodes a glucose 6-phosphate isomerase (PGI, EC 5.3.1.9) responsible for Glc6P to fructose 6-phosphate (Fru6P) conversion, PPi-dependent 6-phosphofructokinase (PFK, EC 2.7.1.11) (Cthe_0347) and ATP-dependent PFK (EC 2.7.1.11) (Cthe_1261) catalyze the conversion of Fru6P to Fru16BP, and Cthe_0349 encodes a fructose-1,6-bisphosphate aldolase (FBA, EC 4.1.2.13) regulating the conversion of fructose 1,6- bisphosphate (Fru16BP) to glyceraldehyde 3-phosphate (GAP), expressed upregulated activities in CGs1 as compared to FAs1, GAs1, FAs8, and GAs8. Gene expression patterns continued their upregulation trend in CGs1, as the three genes Cthe_0707, Cthe_0946 and Cthe_1292 encoding PGAM (EC 3.1.3.3) catalyze the conversion of 3-phosphoglycerate (3PGA) to 2-phosphoglycerate (2PGA) were found to be upregulated in CGs1 relative to FAs1, GAs1 and the evolved strains FAs8 and GAs8. Considering metabolite intermediates data from Nochur et al. [31], transcriptomic and metabolomic data from Yang et al. [21] and our RNA-seq together, we speculate that the low expression levels of EMP pathway genes in FAs1 and GAs1 led to less efficient monosaccharide metabolism, eventually causing the poor growth of *H. thermocellum*.

Regarding the involved strain GAs8, various key genes including Cthe_0347, Cthe_1956, Cthe_1292, Cthe_0143, and Cthe_0217 were found to have increased their mRNA expression to a certain extent when compared to the parent strain, GAs1. In contrast, in FAs8, only two genes (Cthe_1265 and Cthe_0946) exhibited a slightly upregulated pattern relative to FAs1. Therefore, transcriptomic profile alone cannot adequately explain the acquired phenotype of FAs8. Additional information from genome resequencing with genetic mutations may help to elucidate the phenotype of FAs8 better.

#### 3.5.2. Genes Downstream of PEP

The malate shunt is responsible for catalyzing the indirect conversion of PEP to pyruvate via malate. The absence of an annotated pyruvate kinase sequence in the *H. thermocellum* genome, no pyruvate kinase (PYK, EC 2.7.1.40) enzyme activity detected in cell extracts, and pyruvate phosphate dikinase (PPDK, EC 2.7.9.1) is unlikely to be crucial for the conversion of pyruvate from PEP [60,61], leading to an assumption that pyruvate formation could be proceeded predominantly via a malate shunt [62]. Moreover, metabolic flux ratio (METAFoR) analysis also suggested that the malate shunt is the primary pathway which directs the carbon flux from PEP to pyruvate [61]. In our RNA-seq, two of three genes involved in the malate shunt, such as NADP^+^-dependent malic enzyme (ME) (Cthe_0344) and NADH-dependent malate dehydrogenase (MDH) (Cthe_0345), demonstrated elevated expression levels in CGs1, whereas the gene Cthe_2874 encoding phosphoenolpyruvate carboxykinase (GTP), which is responsible for the conversion of PEP to oxaloacetate (OAA), maintained an upregulated level in FAs1, GAs1, FAs8, and GAs8. Furthermore, as ITP can substitute for GTP in the carboxylase reaction, the upregulation of the inosine 5-monophosphate dehydrogenase-encoding gene (Cthe_0681) in CGs1, which is an enzyme involved in the synthesis of guanine nucleotides, further supports the idea that the malate shunt is the main glycolytic route for producing pyruvate in *H. thermocellum* [19]. Alternatively, OAA can be directly decarboxylated into pyruvate using the membrane-bound enzyme oxaloacetate decarboxylase (ODC, EC 4.1.1.3). Although previous studies [23,38] provided evidence of the moderate expression of the ODC gene Clo1313_1523 (~Cthe_0701 in ATCC 27405), our RNA-seq data revealed that the gene Cthe_0701 [1.47/-2.03|2.43/0.78] had higher expression in FAs1 and GAs1, similar to its expression pattern in planktonic cells, a common cell fraction of bacterial population in carbon-depleted medium [22]. Although Olson et al. [62] could not detect ODC activity in their study, we assume that the inconsistency may be due to the different strains of *H. thermocellum* used in these works (e.g., Olson et al. used *H. thermocellum* DSM 1313, whereas we used *H. thermocellum* ATCC 27405).

Gene IDs and corresponding log_2_ fold change (FC) values (number 1/number 2/number 3/number 4) are displayed in the box. Text in red represents expression levels of FAs1 and FAs8, respectively, and text in green displays expression levels of GAs1 and GAs8, respectively. Nomenclature of metabolites is as follows: 2PGA: 2-phosphoglycerate; 3PGA: 3-phosphoglycerate; 13BPGA: 1,3-bisphosphoglycerate; CD ABC transporter: Cellodextrin ATP-binding cassette transporter; Fru: Fructose; Glc: Glucose; CBP: Cellobiose phosphorylase; Fru6P: Fructose 6-phosphate; Fru16BP: Fructose 1,6- bisphosphate; FBA: Fructose-bisphosphate aldolase; GA3P: Glyceraldehyde 3-phosphate; Glc: Glucose; GAP: Glyceraldehyde 3-phosphate; Glc1P: Glucose 1-phosphate; Glc6P: Glucose 6-phosphate; GCK: Glucokinase; Mal: Malate; MDH: Malate dehydrogenase; ME: Malic enzyme; OAA: Oxaloacetate; PGI: Glucose-6-phosphate isomerase; PEP: Phosphoenolpyruvate; PFK: Phosphofructokinase; PGAM: Phosphoglycerate mutase; PPDK: Pyruvate phosphate dikinase; PYK: Pyruvate kinase; PEP: Phosphoenolpyruvate; Pi: Inorganic phosphate.

### 3.6. EMP Pathway

Among the five operons encoding cello-oligosaccharide ABC transport proteins identified by Nataf et al. [63], three operons, CbpA, CbpB, and CbpD, were experimentally proven to be specific to cellodextrins of different lengths (G2–G5). Therefore, it is not expected that these sugar transporters would exhibit high levels of expression during growth on monosaccharides. Due to the absence of cellodextrins in the growth medium, Rydzak et al. [20] could not detect proteins expressed from either the CbpC (Cthe_2125–2128) or CbpD (Cthe_2446–2449) operons in *H. thermocellum* cells. However, high expression levels of these sugar transporter-encoding genes were found during growth on fructose and glucose in this present study. The mRNA expression levels of ATP-binding protein *nbdA* (Cthe_0391), inner membrane translocator *msdA* (Cthe_0392), and ribose ABC transporter *cbpA* (Cthe_0393) were significantly upregulated in FAs1 and GAs1 (Figure 4). Additionally, the transmembrane sugar transport protein *msdB1* (Cthe_1019) in the CbpB operon also exhibited an upregulated expression level. As CbpC transport proteins can bind to glucose [63], it was not surprising that two of four genes in the CbpC operon, namely the binding protein-dependent transport system *msdC1* (Cthe_2125) and permease component *msdC2* (Cthe_2126), were highly expressed in GAs1 and, to a lesser extent, in FAs1. Four genes of the CbpD operon (Cthe_2446–2449), such as ribose ABC transporter *cbpD*, ATP-binding protein *nbdD*, inner-membrane translocator *msdD*, and phosphoglycerate mutase *pgmD*, were upregulated in FAs1 and GAs1. In the study of Rangel and coworkers [64] where xylose was used as the sole carbon source, two sugar transporter genes *cbpC* (Cthe_2128) and *cbpD* (Cthe_2446) were found to be upregulated. Interestingly, in our study, after numerous generations, these genes of the CbpD operon still maintained their highly expressed activities in FAs8 and GAs8, thus indicating their fundamental importance in hexose sugars transport. In contrast, after adaptation to fructose was well established, the expression levels of genes encoding sugar transporters, except for the CbpD operon, significantly decreased in FAs8 and GAs8 (Figure 4).

### 3.7. Energy Generation and Redox Balance

With regards to energy production, while all F-type ATPase genes (Cthe_2602–2609) of FAs1 and GAs1 were downregulated, some V-type ATPase genes were upregulated in GAs1 (Cthe_2266–2269) or in both FAs1 and GAs1 (Cthe_2262–2265). Cellular redox imbalance under certain conditions may cause incomplete substrate utilization and slow growth of bacteria. Redox metabolism of *H. thermocellum* is complex, with multiple reactions that shuttle electrons between reduced ferredoxin (Fd), NADH, and NADPH. These reactions use different enzyme complexes, such as reduced Fd:NAD(P) oxidoreductases and several Fe–Fe and Ni–Fe hydrogenases that are activated by hydrogenase maturase (HydG) [61]. Therefore, *H. thermocellum* produces molecular hydrogen via a hydrogenase-mediated pathway to dispose of the excess reductants generated during carbohydrate catabolism [10]. Hydrogen production has important roles in regenerating oxidized Fd and NADH as it is used as the final electron acceptor for pyruvate:ferredoxin oxidoreductase (PFOR) during the conversion of pyruvate to acetyl-CoA [65]. Sander et al. [5] reported that four redox-active pathways, namely sulfate transport and metabolism, ammonia assimilation, porphyrin biosynthesis, and [Ni-Fe] Fd-dependent hydrogenase, in *H. thermocellum* DSM 1313 showed decreased transcription after the addition of methyl viologen, a redox-active chemical, to balance cellular redox. In the present study, FAs1 exhibited the upregulated expression in [Ni–Fe] hydrogenase (Cthe_3013, Cthe_3016, Cthe_3018, Cthe_3019, Cthe_3020, Cthe_3022, Cthe_3023, and Cthe_3024), ammonia assimilation-related genes (Cthe_0197–0199) and porphyrin biosynthesis (Cthe_2525), but several gene involved in sulfate metabolism (Cthe_2531–2534, Cthe_2536–2538) were downregulated (Appendix A). On the other hand, GAs1 had highly expressed genes in both [Ni–Fe] and [Fe–Fe] hydrogenases (i.e., Cthe_3013, Cthe_3014, Cthe_3018, Cthe_3020, Cthe_0428, Cthe_0429, Cthe_0430), ammonia assimilation genes (Cthe_0197–0199), porphyrin biosynthesis gene (Cthe_2525, Cthe_2527, Cthe_2528, Cthe_2529), and sulfate transport (Cthe_2533). The results suggest that FAs1 and GAs1 did not suffer much from redox imbalance as hydrogen production was favorably maintained for cell redox homeostasis. Transcriptomic data of FAs8 and GAs8 showed that genes related to ammonia assimilation (Cthe_0197–0199) still sustained their elevated expression relative to CGs1. However, most genes involved in [Ni–Fe] hydrogenase, porphyrin biosynthesis, and sulfate metabolism in FAs8 and GAs8 were downregulated in comparison with their respective FAs1 and GAs1. The downregulation pattern may indicate that the evolved phenotypes do not need a large amount of hydrogen to function as an electron acceptor in rebalancing the cells redox state, however, further investigation is needed to thoroughly clarify this phenomenon.

### 3.8. Other DEGs in Different Categories

Under conditions offering only poorly fermentable substrates, the bacterium needs to increase its motility and signal transduction systems to seek a better carbon source in the environment. This explains the greater numbers of DEGs in category N and category T in FAs1 and GAs1 relative to CGs1 (Appendix A). However, this upregulated pattern of genes in category N was altered in FAs8 and GAs8 when the adaptation to hexoses was well established. NAD^+^ and NADP^+^ are important cofactors that are usually involved in catabolism and anabolism, respectively [22]. In FAs1 and GAs1, the four genes responsible for NAD^+^ biosynthesis via L-aspartate (Cthe_0325, Cthe_2355, Cthe_2356, Cthe_1241) and one gene ATP-NAD kinase (Cthe_0816) for the conversion of NAD^+^ to NADP^+^ were downregulated relative to CGs1. This supports the observation that CGs1 had a higher rate of carbohydrate assimilation and faster cell growth.

Regarding the antioxidant response, a rubredoxin-type Fe(Cys)4 protein gene (Cthe_0063) was found to be upregulated in GAs1. Another rubredoxin protein gene (Cthe_2164) had upregulated expression in both FAs1 and GAs1. This class of protein is strictly found in anaerobic bacteria and archaea where it plays roles in the reduction of oxygen. Other antioxidant protein FAD-dependent oxidoreductase gene Cthe_0200 exhibited elevated expression in GAs1 and FAs8, whereas another FAD-dependent oxidoreductase gene (Cthe_0560) was upregulated in both. Additionally, a peroxiredoxin gene (Cthe_1965) was upregulated in GAs1 and GAs8. In general, just some antioxidant-related genes were statistically significantly expressed in carbon-challenged medium. Other downregulated genes such as glutathione S-transferase N-terminal domain (Cthe_0235) were found in FAs1, GAs1, FAs8, and GAs8, thioredoxin (Cthe_0360) in FAs1 and FAs8, peroxiredoxin (Cthe_1465) in FAs1, GAs1, and FAs8, and FAD-dependent pyridine nucleotide-disulphide oxidoreductase (Cthe_1164) in FAs8.

In general, there was no significant difference between quantities of DEGs in the parent strains FAs1 and GAs1 and in the evolved FAs8 and GAs8. However, the ratio of upregulated genes to downregulated genes was tremendously decreased from the parent strains to the evolved strains. We assume that in a growth medium with hexose sugar as a sole carbon source and energy, FAs1 and GAs1 cells needed to exploit all necessary pathways to produce energy, transport organic and inorganic elements, induce signal transmission, and accelerate transcriptional and translational processes to cope with the harsh conditions. This explains why numerous genes in all categories were upregulated in comparison with CGs1, except for genes involved in EMP pathway. After the useful mutations for growth in fructose- and glucose-based medium had been well established, their expression levels were repressed to save energy for other essential physio-biochemical aspects. Consequently, the evolved strains achieved greater biomass, higher growth rate, and higher ethanol yield. Furthermore, the significant upregulation of urease (Cthe_1812–18) and urea ABC transport (Cthe_1819–23) genes in FAs8 was recorded, whereas most nitrogen assimilation genes in GAs8 were downregulated, except for Cthe_1823 (Appendix A), which may explain the higher biomass of FAs8 as compared to that of GAs8.

### 3.9. Genomic Analysis

To overcome environmental stress, the mutation rate in the bacterial genome in the late phase of the adaptation period increases to allow the development of an evolved phenotype [36]. In the present study, which required the bacteria to adapt quickly to harsh nutritional conditions, various mutations were detected in *H. thermocellum* using genomic resequencing. In the study by Yayo et al. [38] *H. thermocellum* DSM 1313 was cultured in the carbon-limited chemostats with an increasing glucose or fructose concentration and decreasing cellobiose concentration strategy, in our study, we cultured *H. thermocellum* ATCC 27,405 in batch culture with sole fructose-, glucose-, or cellobiose-supplemented medium from the beginning of culture. The addition of 100% less-preferred hexose sugars to growth medium created a harsh condition and nutritional stress for *H. thermocellum*. Consequently, the only way for the bacterium to survive and thrive was to quickly adapt to new carbon sources by mutation acquisition.

In the genomes of the evolved strains FAs8 and GAs8, we found an insertion or deletion (indel) mutation in the *MDH* (Cthe_0345) gene, an insertion/deletion (indel) & stop-gain mutation in the *GCK* (Cthe_0390) gene and a nonsynonymous (non-syn) mutation in the *CBP* (Cthe_1221) gene (Appendix A). While the deletion of a putative ROK family transcriptional regulator Clo1313_1831 (~ Cthe_0390 in strain ATCC 27405) facilitated immediate growth on glucose and a reduced lag phase on fructose [38], in our study, an insertion of T at the 49/1215 position caused a frameshift and stop gain in the evolved strains FAs8 and GAs8. This type of mutation may lead to a severe disruption in the transcriptional process of the Cthe_0390 gene, thus accounting for its low expression level. We may propose that this mutation also leads to the reduced lag phase on fructose and glucose, as observed by Yayo et al. [38].

We also found various mutations in sugar transporter genes of the evolved strains. Specifically, an indel mutation (deletion of GG at position 19/1050) was observed in the *msdA* gene (Cthe_0392) of FAs8 and GAs8. Both indel and stop-gain events occurred in the *nbdA* gene (Cthe_0391) of FAs8. The *cbpB* gene (Cthe_1020) in GAs8 had a non-syn mutation (Ala was replaced by Val^92^). The gene encoding sugar-binding periplasmic protein *cbpC* (Cthe_2128) of FAs8 and GAs8 had a non-syn substitution (Gln was altered by Glu^355^). Since the sugar-binding protein CbpC operon could bind glucose and cellobiose, we suspect that this mutation might improve the binding affinity of this protein to glucose and fructose. We also found a stop-gain mutation (Gln148*) in the *nbdD* gene (Cthe_2447) of FAs8. This type of mutation may abolish the function of the *nbdD* gene, however, other transporter(s) may compensate for its role, as observed in the study of Rangel et al. [64]. Since these genes belong to the CbpA, CbpB, CbpC, and CbpD operons, which are responsible for polysaccharide G2–G5 transport, the mutations found in the evolved strains might improve the transport of Fru or Glc into the cells by altering substrate specificity. A non-syn SNP (G538A) may disrupt the gene function of the canonical master sporulation regulator *spo0A* (Cthe_3087), which may result in faster growth of the evolved strains FAs8 and GAs8. The mutation of *spo0A* was consistent with its downregulated regulation in FAs8 and GAs8. Our finding was corroborated by the study of Linville and coworkers [8,13] as a stop codon occurred in the coding sequence (CDS) of Clo1313_0637 (~Cthe_3087) of a mutant strain accounted for a better tolerance to and a faster growth in Populus hydrolysate-containing medium.

Regarding cellulosomal genes, we found a synonymous (syn) mutation in *olpB* (Cthe_3078) gene in the evolved FAs8, as A was replaced by G^5403^_,_ which may cause a reduction in gene expression. Furthermore, a syn mutation (Ala^313^Val) was acquired in *htpG* (Cthe_0550) gene of FAs8 and GAs8. The *htpG* gene was found to be upregulated in all stress treatments, including ethanol stress, furfural, or heat shock, which points out the important role of this gene in stress tolerance [11]. Although this is just a syn mutation, its importance should be noted, as the ecological adaptation of bacteria to a new environment can also occur through synonymous changes, as they affect the nature of mRNA/tRNA interactions [66]. In addition, codon selection is common in both prokaryotes and eukaryotes and fundamentally influences the synthesis of a particular polypeptide and/or the accuracy of translation [67]. UvrABC excinuclease plays an important role in bacterial nucleotide excision repair [68]. In the present study, a non-syn mutation was found in the open reading frame of the excinuclease ABC subunit *uvrB* gene (Cthe_0309), where amino acid His was replaced by Arg at position 402 (Appendix A). An indel mutation (deletion of Ala at position 192 causes a frameshift variant) occurred in the *deoR* transcriptional regulator (Cthe_2441) of the evolved FAs8. In *E. coli*, *deoT*, a DeoR-type transcriptional regulator, represses the expression of genes involved in a variety of metabolic pathways related to maltose, fatty acid β-oxidation and peptide degradation [69]. In FAs8 and GAs8, *araC* (Cthe_3164) acquired a stop mutation at position 227 (this gene is homologous to the *araC* gene in *E. coli*, with 22.67% identity). In *E. coli*, *araC* functions as a transcription factor regulating the expression of several genes involved in the transport and metabolism of L-arabinose [70], and regulators of the AraC/XylS family are involved in the metabolism of certain carbon sources [71].

It is worthy to note that, although the isolation of individual evolved strains is necessary for identifying causative mutations and for further genetic engineering [37], in our pangenome analysis, most sequencing reads associated with the mutated genes had SNVs, which confirm that all bacterial cells in the evolved populations already acquired mutations which are beneficial to hexoses-adapted phenotypes.

## 4. Conclusions

In summary, the advances in NGS and RNA-seq technologies allowed us to gain insight into the genetic basis and dynamics of adaptation in bacterial populations. RNA-seq showed large changes in core metabolic pathways during growth on different carbon sources, and the most significant transcriptional level differences were related to the EMP pathway genes, ABC sugar transporters and CAZymes. On the other hand, genomic analysis pointed out that mutations in various genes related to carbon transport (ABC transporters) and carbon metabolism occurred in the genomes of the evolved FAs8 and GAs8 strains. These genes could be good candidates for further metabolic engineering approaches to improve biofuel production using this bacterium. This study also has enhanced our understanding of the physiology, metabolism, and nutritional adaptation of *H. thermocellum*.

## Figures and Tables

**Figure 1 microorganisms-09-01445-f001:**
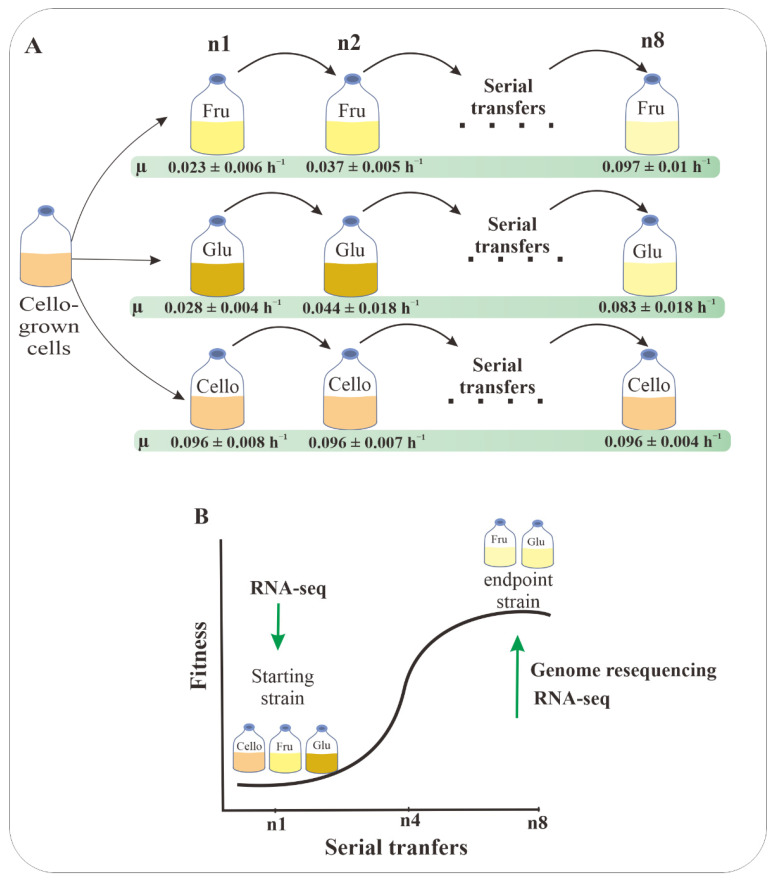
Schematic diagram of the present study. (**A**) Adaptive laboratory evolution was applied to FAs1 and GAs1 to improve their growth on fructose and glucose, respectively. After ~40 generations, growth rates of FAs8 and GAs8 were dramatically increased. (**B**) Starting strains FAs1, GAs1, and CGs1 at the late stationary phase were used for RNA-seq and the end-point strains FAs8, GAs8 were used for genomic analysis and RNA-seq.

**Figure 2 microorganisms-09-01445-f002:**
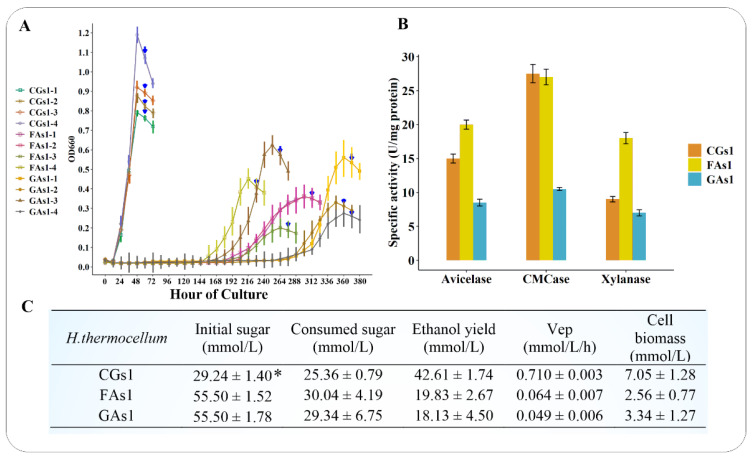
*H. thermocellum* growth patterns, cellulosomal enzyme activity, sugar consumption, and ethanol fermentation. (**A**) Growth profiles of CGs1, FAs1, and GAs1 in four biological replicates. The blue arrowheads indicate the sampling points used for RNA-seq, RT-qPCR, and cellulosome isolation. (**B**) Specific enzyme activity of the *H. thermocellum* cellulosome. (**C**) Sugar consumption, ethanol yield, volumetric ethanol productivity, and cell biomass of CGs1, FAs1, and GAs1. All data in Figure 2B,C represent the mean ± sd of biological triplicate. Note: * mmol glucose equivalents; Vep: Volumetric ethanol productivity (mmol/L/h).

**Figure 3 microorganisms-09-01445-f003:**
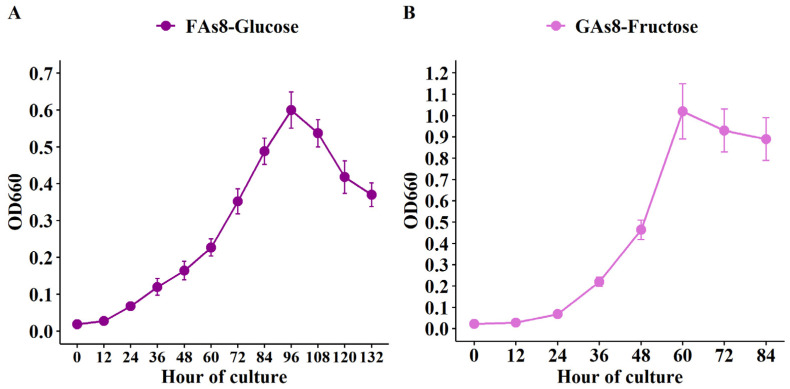
Growth pattern of the involved strains on different hexose sugars. (**A**) Growth pattern of FAs8 on fresh glucose medium. (**B**) Growth pattern of GAs8 on fresh fructose medium.

**Figure 4 microorganisms-09-01445-f004:**
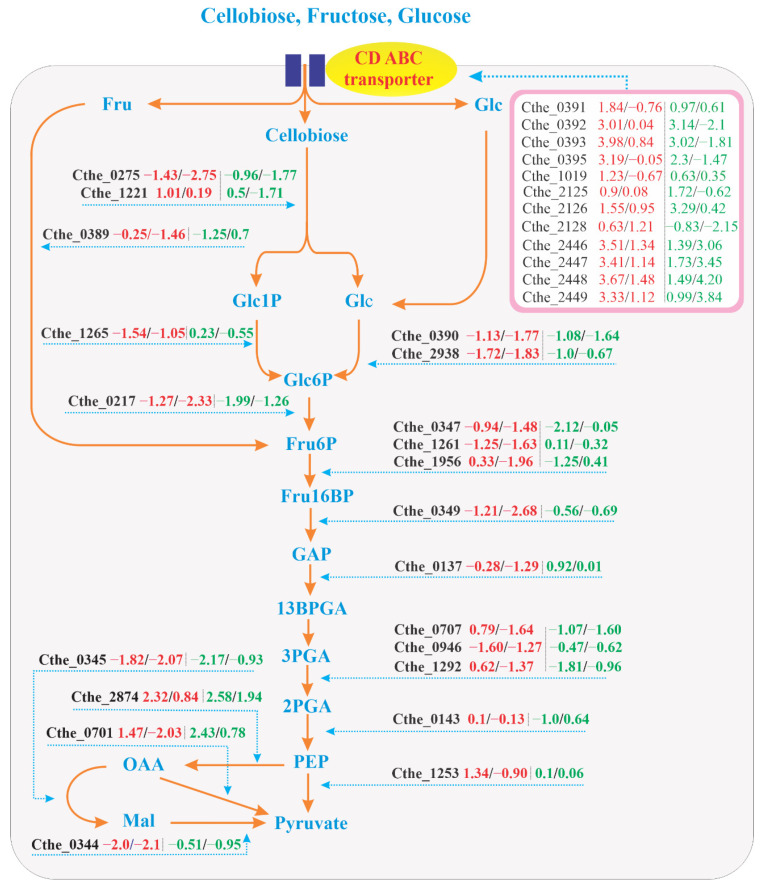
Diagram of the genes involved in EMP pathway in FAs1, FAs8, GAs1, and GAs8, relative to CGs1.

**Table 1 microorganisms-09-01445-t001:** Hexose sugars consumption, ethanol and CO_2_ production, cell biomass and ethanol yield after seven sequential transfer events.

Fructose-Adapted Cells	Consumed Sugar (mmol/L)	Ethanol (mmol/L)	CO_2_ (mmol/L)	Cell Biomass (mmol/L)	Ethanol Yield (mmol Eth/mmol Sugar)
FAs1	30.04 ± 4.19	19.83 ± 2.67	3.24 ± 0.46	2.56 ± 0.77	0.66 ± 0.03
FAs2	15.24 ± 1.67	7.610 ± 1.13	1.79 ± 0.24	3.68 ± 1.24	0.50 ± 0.02
FAs3	17.94 ± 2.63	8.110 ± 1.68	2.14 ± 0.28	5.22 ± 2.4	0.45 ± 0.02
FAs4	18.22 ± 2.80	8.260 ± 1.36	2.09 ± 0.31	5.54 ± 2.43	0.45 ± 0.03
FAs5	22.58 ± 1.59	11.30 ± 1.21	2.74 ± 0.22	6.04 ± 1.12	0.50 ± 0.02
FAs6	26.21 ± 2.50	11.97 ± 0.98	2.83 ± 0.14	8.33 ± 1.14	0.46 ± 0.04
FAs7	30.37 ± 1.84	14.45 ± 1.24	2.98 ± 0.28	9.32 ± 0.83	0.48 ± 0.05
FAs8	34.23 ± 2.74	17.54 ± 1.06	3.68 ± 0.29	9.77 ± 1.37	0.51 ± 0.06
**Glucose-Adapted Cells**					
GAs1	29.34 ± 6.75	18.13 ± 4.50	4.42 ± 1.10	3.34 ± 1.27	0.62 ± 0.04
GAs2	17.19 ± 2.23	8.800 ± 3.20	2.85 ± 1.25	3.75 ± 1.83	0.51 ± 0.02
GAs3	18.51 ± 2.92	9.130 ± 1.20	3.03 ± 0.70	4.48 ± 0.66	0.49 ± 0.03
GAs4	21.05 ± 2.12	9.910 ± 1.72	3.15 ± 0.80	5.87 ± 2.36	0.47 ± 0.04
GAs5	21.52 ± 3.47	10.00 ± 1.30	3.24 ± 0.49	6.30 ± 1.73	0.46 ± 0.04
GAs6	23.41 ± 2.50	11.83 ± 1.23	3.29 ± 0.42	6.18 ± 2.25	0.51 ± 0.02
GAs7	23.76 ± 3.50	12.28 ± 3.85	3.35 ± 0.82	6.13 ± 2.86	0.52 ± 0.06
GAs8	24.88 ± 2.74	12.35 ± 1.18	3.41 ± 0.41	7.27 ± 4.49	0.50 ± 0.05

Note: The results were calculated and displayed as averages of four biological replicates.

## Data Availability

The RNA-seq raw counts could be found in the Appendix A of this paper.

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
