# Peer review of "Utilization of Monosaccharides by *Hungateiclostridium thermocellum* ATCC 27405 through Adaptive Evolution"

_microorganisms, 2021, doi:10.3390/microorganisms9071445_

Round 1
Reviewer 1 Report
The authors have addressed most of my comments in their rebuttal and significantly improved parts of the text. Although some of my comments appeared to be overly critical, I still feel some parts of the text should be improved.
1. In their rebuttal the authors clearly describe why the use of glucose and fructose is relevant. In my opinion they should add similar wording to the Introduction to make this clear to all readers: "The use of glucose in this study is based on the fact that the accumulation of glucose at the end of substrate fermentation was noticed in a previous study [1] and the presence of glucose in medium, in turn, inhibits catalytic activity of cellulosomes [2]. Accordingly, improving the utilization of glucose by H. thermocellum is beneficial for cellulosic bioethanol production. Although the chemical formula of glucose and fructose is the same, differences in their molecular formations leads to changes in cellulase production of glucose- and fructose-adapted cells. This phenomenon was first observed by Johnson et al. [3] decades ago, however, it has not been elucidated on a point of view of mRNA expression level."
2. The Results and Discussion sections starts in the middle of an experiment without introduction of what was done to get to this stage and why it was done. A few lines will do and make it so much easier for the reader.
Figure 1 is Figure 2 and the actual Figure 1 has no legend to it.
For the actual Figure 1 (IU/mg) has no definition for what mg refers to. The authors have answered my question on this in their rebuttal, but did not add the information (mg of protein of the extracted cellulosomes) to the text. Please add this to make it clear for the reader.
3. It is still not clear to me why the authors use 10 g/L for glucose or fructose and 5 g/L for cellobiose. They state that cellobiose is a disaccharide, but that is only relevant if they have a concentration in mmol/L and not in g/L. Now they have double the amount of sugar input for the glucose and fructose lines. In the end, the consumption seems to be the similar.
Minor edits:
Throughout the text H. thermocellum should be in italics
Throughout the text when talking about genes the names should not be with capitals and should be in italics, e.g.: Line 502 the exoglucanase gene celS , scaffoldin protein gene cipA, and two anchoring protein genes olpB and orf2p were affected by...
Line 130 delete "was used"
Line 439 "From the 1st the 7th" should read "From the 1st to the 7th"
Reviewer 2 Report
The authors have made several changes to the manuscript that improve the quality, though my primary concerns have not been addressed in this revision. This lack of responsiveness to my concerns is exacerbated by the lack of a response letter from the authors to explain where and why they did or did not change the manuscript. As a result, I still cannot recommend publication without minor revisions.
In particular, the authors have not clarified how the transfers were carried out (e.g., are FAs2-1, 2-2, and 2-3 prepared from FAs1-1, 1-2, and 1-3, respectively, or from pooled FAs1-1, 1-2, and 1-3?) and have removed Figures 3 and 4 of the original manuscript rather than discuss them in more detail. It appears they have replaced those Figures with Table 1, which masks much of the variability observable in those figures in the original manuscript.
Additionally, the panel with the growth curves in the new Figure 1 is not easy to read, and even here there is significant variation across series, at least for the FA and GA series.
Author Response
Please see the attachement.

This manuscript is a resubmission of an earlier submission. The following is a list of the peer review reports and author responses from that submission.
Round 1
Reviewer 1 Report
The authors present their results on a genomic characterization of Hungateiclostridium thermocellum grown on cellulose or adapted to glucose and fructose. Their analyses include growth curves, carbohydrolase assays, transcriptomics and resequencing of adapted strains. Although the manuscript is generally easy to read, there are some important aspects that need additional explanation. From the introduction it is not clear to me why fructose utilization would be a relevant topic to study. I see no apparent connection with lignocellulosic or cellulosic substrates that are typically associated with this species. If inverted sucrose would be the rationale, then it should be stated and also sucrose itself as a sugar would be relevant to discuss. With regards to lignocellulosic substrates the utilization of arabinose, galactose, rhamnose, and xylose, next to glucose, seem more relevant than fructose. The results of the transcriptomics studies mainly confirm results from other studies and add little new insights. The main observation is the lower expression of several glycolytic genes in the EMP pathway in glucose- and fructose-grown cells. However, the authors do not provide transcriptomics data on the 8th transfer series to see if this has changed in the adapted strains with better growth. In my opinion these data should be added to the study. The adaptive evolution (although 8 transfers is not very much in this respect) and resequencing of adapted strains is interesting, but restricted to speculation. Validation of the impact of specific mutations would be required, although I realize the practical challenge associated with that. Alternatively, sequencing of mutant strains from a series of independent adaptation lines would give a better insight in essential modifications for improved sugar utilization. All together, I feel that the study involves a significant amount of work, but remains too descriptive and provides too little new insights in its current form.
Minor comments:
Reference could be made to the work of Tafur Rangel et al. (doi: 10.1038/s41598-020-71428-6) on the transcriptional comparison of cellobiose and xylose-grown H. thermocellum.
Lines 58-60. FAs1, GAs1 and CGs1. From Figure 3 I understand that s1 refers to series 1 (and s8 to series 8). It would be nice for the reader if this is mentioned in the text a bit earlier than in line 486.
Lines 89-92. What is the rationale for using 10 g/L for glucose and fructose and 5 g/L for cellobiose?
Line 95. I would expect the recipe of GS-2 medium straight after this sentence instead of at the end of this paragraph (lines 108-119).
Line 268. CGs1 should read GCs1?
Line 281. specific enzyme activity is in IU per mg of what?
Line 284. It would be informative to include a column with remaining sugar. Fermentations start with ~55mM glucose and 14 mM cellobiose (28 mM in glucose equivalents). As questioned above, why did you not use the same amounts of initial sugar for the three substrates?
Reviewer 2 Report
This paper describes the genetic adaptation of Hungateiclostridium thermocellum ATCC 27405 to use fructose and glucose as carbon sources (compared to cellobiose as a preferred carbon source) by directed evolution through eight transfer events. The authors discuss several genes and gene classes that are up- and down-regulated through these transfer events and the implications for engineering this strain for biofuel (ethanol) production. The manuscript is generally well-written, the experimental plan and results are logical, and the content is likely of interest to Microorganisms readers. I recommend publication after minor revisions.
There are two issues that need further description. The first is how the transfers were carried out. For example, are FAs1-1, FAs1-2, and FAs1-3 the biological replicates? And from this step, how are FAs2-1, FAs2-2, and FAs2-3 prepared? Was one replicate from the FAs1 generation transferred to three separate cultures for FAs2, or did FAs1-1 become FAs2-1 after transfer, FAs1-2 become FAs2-2, etc.?
Similarly, the growth curves in Figures 3 and 4 should be discussed in more detail. There seem to be significant differences in growth between the tubes in a given generation, especially in Figure 4. What leads to this variation and what are the implications of it?
Additionally, inclusion of some figures from the supporting info in the manuscript would assist the reader to follow the discussion around the gene upregulation and downregulation. In particular, Figures S1, S5, and S6 would be beneficial additions to the manuscript text. The caption labels for Figures S5 and S6 also appear to be switched.
Other minor issues: Figure 1: should the arrowheads be the same color as data? (similar comment for Figures 3 and 4.) In the Figure 1 caption: FAS1 (vs. FAs1). How is mmol/L of cell mass defined?